# Repeated Massage Improves Swimmers’ Perceptions during Training Sessions but Not Sprint and Functional Performance: A Randomized Controlled Trial

**DOI:** 10.3390/ijerph20031677

**Published:** 2023-01-17

**Authors:** Flávia A. Carvalho, Natanael P. Batista, Fernanda P. Diniz, Aryane F. Machado, Jéssica K. Micheletti, Carlos M. Pastre

**Affiliations:** Department of Physiotherapy, Sao Paulo State University (UNESP), Presidente Prudente 19060-900, Brazil

**Keywords:** muscle soreness, massage therapy, recovery, swimming

## Abstract

This study aimed to investigate the effects of repeated massage adjusted for swimmers’ training on the perceptive, functional, and performance outcomes of a sprint. We also investigated the effects of a single short massage on swimmers’ self-reported perceptions after resistance training. This cross-over randomized controlled trial with concealed allocation, assessor blinding, and intention-to-treat analysis included 19 male and female competitive swimmers between 12 and 20 years old. Participants were subjected to three 12-min interventions over a week between resistance and swim training and monitored regarding training load and perceptions. After the intervention week we assessed: perceptive (well-being, heaviness, tiredness, discomfort, and pain), performance (sprint time, FINA points, and stroke characteristics), and functional outcomes (flexibility, squat jump, bench press, proprioception), in addition to athlete beliefs and preferences. A massage was defined as consisting of sliding movements on the arms, back, and anterior thigh, with metronomic rhythm control (1:1), and was divided into two protocols: superficial massage (SM) (light touch) and deep massage (DM) (light, moderate, intense effleurage) while the control (CON) rested. After repeated massage (SM and SM), participants had less chances to report tiredness, and they also maintained perceptions of well-being while CON got worse throughout the week. However, we found evidence of worsening of the perceptions of heaviness and pain at the main stages of the swim training for the massage groups. SM and DM had no effects over sprint and functional performance. Our results suggest that the swimmers were able to train harder with no harm to recovery.

## 1. Introduction

In the XXIX Olympic games of Beijing, swimming had one of the lowest competition injury rates but a high incidence of overtraining injuries [1]. Nevertheless, Pollock et al. found that almost half of the swimming coaches interviewed for their study did not implement individualized recovery strategies or follow recommendations for reducing overtraining injuries [2]. According to Meeusen et al., the main cause of these types of injuries is the lack of balance between training and recovery [3], and therefore, recovery elements should be introduced to the periodization to match specific recovery needs and enhance adaptation [4]. 

Post-exercise recovery is a restorative process relative to time, involving different aspects such as physiological and psychological variables [5]. To implement post-exercise recovery strategies, it is necessary to understand training characteristics and the type of stress; therefore, load monitoring is essential for understanding an athlete’s experiences over training sessions. Marcora et al. [6] found that athletes instinctively regulate the intensity of exercise according to how they feel; in this context, complaints of previous resistance training can influence subsequent swim training.

Massage is one of the most widely used recovery strategies in international competitive events [7] and has evidence of reducing the delayed onset of muscle soreness (DOMS) [8,9], stiffness [10], and immune compounds that may impact fatigue [11]. Athletes interviewed by Kennedy et al. reported that only one session of massage is not enough for recovery [12], however, some studies [8,13,14] showed that repeated massage was not superior to a single application. This is possibly because of the natural regression to baseline, since there was not any type of exercise between sessions. A meta-analysis by Poppendieck et al. found small effects on performance after short massages with better results in the short term after intensive mixed exercise [15]. Only one study investigated post-exercise massage in swimmers and found significant effects on lactate concentration and sprint time [16]. 

Considering swimmers’ perceptions, their training context, and the current evidence, we hypothesize that massage favors acute recovery from resistance training, enabling adequate swimming and, consequently, improving performance. Therefore, this study aimed to investigate the effects of repeated massage adjusted to a swimmer’s training on perceptive, functional, and performance outcomes and the effects of a single massage on swimmers’ self-reported perceptions after resistance training. We hypothesize that massage would recover swimmers’ perceptions shortly after intervention improving their training abilities which could, in turn, result in positive repercussions in sprint and functional performance at the end of the week.

## 2. Materials and Methods

### 2.1. Study Design

A single-blinded crossover randomized controlled trial with 1:1:1 allocation was reported according to the Consolidated Standards of Reporting Trials (CONSORT) checklist (Appendix A). Identification, competitive level, age, height, and weight were obtained at baseline. An independent researcher performed balanced-block randomization by sorting using the software Excel, with a balanced sex, age, and competitive level ratio, and allocated the participants to three different sequences of interventions which included (i) control, (ii) superficial massage, and (iii) deep massage. Participants undertook a training week followed by an intervention week three times with one week wash-out (Figure 1). Anthropometric, perceptive, functional, and performance data were obtained at testing sessions before and after the intervention week. Perceptive outcomes were also collected over the week at training sessions, which consisted of resistance training adapted from Morais et al. [17], followed by swim training with four stages: warm-up, skills sets, main sets, and cool-down. All activities were performed during the same mesocycle (specific phase), and training loads were monitored.

The participants were assigned at the beginning of each intervention week by an independent researcher to guarantee concealed allocation. Thus, therapists and participants had no previous knowledge of the intervention sequence, however, therapists and participants were not blinded to the intervention. The assessments were performed by independent investigators blinded to group allocation. All randomized participants were analyzed in the groups to which they were randomly assigned, regardless of their adherence to the protocol. However, if participants missed one or both testing sessions, they were excluded from the analysis because testing sessions could only take place once a week due to the athletes’ training routine.

The application time-point was chosen based on a pilot study that found that a 24-h rest after swim training can return perceptions to baseline, but the most inconvenient perceptions appeared after resistance training and higher loads on Mondays, Tuesdays, and Fridays.

### 2.2. Subjects

A convenience sample of athletes was screened from a local swimming team. To be included, male and female athletes should be healthy, over 12 years old, and train regularly on the competitive team. The study was registered on ClinicalTrials (NCT03886376) and approved by the Sao Paulo State University Research Ethics Committee (CAAE: 92348518.2.0000.5402). Both participants and their legal guardians were informed about the study’s risks and benefits and gave informed consent.

### 2.3. Procedures

All procedures were performed at a local sports club with an indoor swimming pool. Training load was monitored by training volume (distance swam in meters) and Session Rating of Perceived Exertion (sRPE) [18] which was obtained by the product of the session duration in minutes and the perceived effort (RPE on a 1–10 scale) for resistance and swim training (over 20–30 min after the main sets), and then summed. 

### 2.4. Interventions

After the resistance training, participants headed to the massage table where they received 12 min of superficial massage (SM) or deep massage (DM) by trained physiotherapists on the arms, back, and anterior thigh with sliding movements controlled by metronome (1:1). Massage lasted 1.5 min on each arm, 1 min on each back trajectory (Appendix B), and 3 min on each thigh. The SM protocol consisted of 12 min of light touch while for DM protocol light, moderate and intense sliding were equally distributed. The participants allocated to the control group (CON) were instructed to rest for 12 min. They were free to sit, stand, or walk by the pool to simulate the actual training scenario, but were instructed not to enter the water or engage in physical activity.

### 2.5. Outcomes

Perceptions were identified by a pilot study that investigated the swimmers’ specific reports after a strenuous training session and then assembled into a short questionnaire. It consisted of five items (well-being, heaviness, tiredness, discomfort, and pain) rated by a Likert scale from 1–5 (nothing, a little, moderate, a lot, and extremely), and was used both on training and testing sessions.

Sprint times were obtained by the coach with a digital chronometer during a 100-m freestyle sprint in a semi-Olympic pool. From this data, we calculated FINA points, an international scoring system to characterize a swimmer’s performance that varies from 0 to 1000. The score was adjusted to an individual’s performance and was calculated as Equation (1), where B is the participant’s best performance on the latest season and T is the sprint time.
FINA Points = 1000 × (B/T)^3^(1)

Stroke characteristics were derived from the number of strokes (n) and time (t) to swim the central 10 m (d) of the swimming pool. These parameters were assessed by trained researchers that were blinded to the assigned interventions. Swimming velocity (SV), stroke frequency (SF), distance per stroke (DPS), and stroke index (SI) were calculated as follows [19]:SV = d/t(2)
SF = n/t(3)
DPS = SV/SF(4)
SI = SV × DPS(5)

Flexibility was assessed by the sit-and-reach (SR) test [20], in which participants were asked to sit with feet touching the SR box and then slide with the dominant hand on top of the other at a maximum distance without flexing their knees. The best score in cm of three attempts with 30-s intervals was registered.

Lower limb power was assessed by the squat jump test [21], in which participants were asked to jump from a semi-squat position without countermovement on a resistive platform (Multisprint, Hidrofit, Brazil). Participants were required to land at the same point of takeoff and rebound with straight legs when landing to avoid knee bending and to keep both hands on the hips throughout the test. The best score in cm of three attempts with 30-s intervals was registered. 

Upper limb power was assessed by the bench press by an isoinertial dynamometer (T-force System, Ergotech, Murcia, Spain) attached to a free bar with a fixed weight of 10 kg. Participants laid on a bench with both feet on the ground, lowered the bar slowly to the chest, and then explosively pressed to full arm extension. They were required to keep the head, shoulders, and buttocks in contact with the bench throughout the lift. The test was performed according to the equipment instructions (http://www.tforcesystem.com/tutorial.php, accessed on 2 February 2018). The best mean propulsive velocity score in m/s of three attempts with 30-s intervals was registered. 

Shoulder proprioception was assessed by an adaptation of the laser pointer-assisted angle reproduction test [22]. Participants were asked to stand one meter from a board fixed on the opposite wall with three targets marked individually at 55°, 90°, and 125° of shoulder flexion. A laser pointer was fixed below the deltoid of the dominant arm with a Velcro strap. The participant was required to point the laser at the targets, memorize the three joint positions, and then reproduce them as before in a randomized order, but with eyes covered. The distance (d) in cm from the target on both the vertical and horizontal axis were transformed in degrees by custom software using Equation (6), and the angular deviations (AD) were obtained by Equation (7). The smaller angular deviation in degrees was registered.
d = 100 × tanZ(6)
AD = √(ZY)^2^ + (ZY)^2^(7)

The trial stopped at the end of seven weeks when participants completed the three interventions. Before and after the trial the participants were asked if they believed massage would be/was effective to improve performance in swim training. After the trial, they were told about the existence of two different massage protocols, and then asked with which intervention they felt more recovered. 

### 2.6. Statistical Analyses

Statistical analyses were performed on SPSS software version 18 (SPSS Inc., Chicago, IL, USA). Effects of massage on sprint and functional performance were analyzed by Generalized Linear Mixed Model with Gamma distribution, and cumulative logit link function, whereas perceptive outcomes were analyzed by Generalized Estimating Equations with ordinal distribution and cumulative logit link function. Dependent variables were rated by a 5-point Likert scale, and the first category (nothing) was used as a reference unless otherwise stated. Intervention, group, and training time points were used as predictors in the repeated massage models, and for the single massage model, the weekday was also considered. Bonferroni adjustments were used for all significant main effects. Parameter estimates (B) and Exp(B) were reported along with 95% confidence intervals, and descriptive data were reported as means and standard deviation (SD). Pearson’s test was used to explore the correlation between training load and performance, and interpreted as small (0.00–0.25), fair (0.26–0.50), moderate to good (0.51–0.75), and excellent (>0.75) [23]. All analyses assumed a level of significance of *p* < 0.05.

## 3. Results

Nineteen swimmers were included in this study (Figure 2). Anthropometric characteristics at baseline are described in Table 1.

The effects of repeated massage were investigated at testing sessions after one week of interventions by their perceptions during a 100-m freestyle sprint, their sprint performance, and functional tests. Our results show no main effects on sprint and functional performance outcomes after repeated massages. Table 2 shows descriptive and mean differences for sprint and functional performance outcomes that were assessed after the sprint. Regarding perceptive outcomes assessed throughout the sprint, we found only time effects showing that participants reported worsening of all perceptions at the middle, end, and after the sprint regardless of the intervention (Figure 3). 

Although there were no effects after one week of intervention, we found interesting changes after single massages during the studied swim training sessions for pain, heaviness, well-being, and tiredness.
Pain

Both massage groups had fewer chances of reporting no pain at the main series on the third day compared to the first (SM Exp(B) = 0.09 CI 95% = 0.02 to 0.35 *p* < 0.01, DM Exp(B) = 0.27 CI 95% = 0.07 to 1.05 *p* = 0.05), whereas CON had more chances of reporting no pain at the first stages of the swim training on the third day compared to the first (during swim warm-up, Exp(B) = 3,25 CI 95% = 1.04 to 10.23 *p* = 0.04, and after swim warm-up, Exp(B) = 4.74 CI 95% = 1.27 to 17.65 *p* = 0.02) (Figure 4A).
Heaviness

DM has more chances of reporting extreme heaviness after the main series on the second day compared to the first (Exp(B) = 6.71 CI 95% = 2.35 to 19.13 *p* < 0.01) (Figure 4B).
Well-being

CON had fewer chances of reporting extremely well until the end of swim training on the third day compared to the first (after resistance training Exp(B) = 0.43 CI 95% = 0.19 to 0.96 *p* = 0.04, after intervention Exp(B) = 0.34 CI 95% = 0.15 to 0.81 *p* = 0.01, during swim warm-up Exp(B) = 0.18 CI 95% = 0.08 to 0.42 *p* < 0.01, after swim training Exp(B) = 0.22 CI 95% = 0.07 to 0.70 *p* = 0.01). On the other hand, both massage groups did not report important changes in well-being throughout the week (Figure 4C).
Tiredness

SM had fewer chances of reporting extreme tiredness at the first stages of the swim training on the third day compared to the first (during swimming warm-up Exp(B) = 0.17 CI 95% = 0.03 to 0.87 *p* = 0.03, and after swimming warm-up Exp(B) = 0.21 CI 95% = 0.06 to 0.79 *p* = 0.02). DM had fewer chances of reporting extreme tiredness after intervention (Exp(B) = 0.25 CI 95% = 0.07 to 0.92 *p* = 0.038) on the third day compared to the first (Figure 4D).

Four participants were not sure, whereas fifteen participants already believed in the effects of massage before the trial, and this proportion did not change after study completion. Moreover, all athletes preferred one of the massage protocols over the control, only two preferred SM over DM due to the intensity, and no unintended effects were observed in any of the groups. Because of a competitive event, the training load of one week was significantly reduced. Nonetheless, this happened equally for all groups and Pearson’s correlation showed no important influence of training load over performance (Sprint time r^2^ = 0.01 *p* = 0.16; FINA Points r^2^= 0.004 *p* = 0.81).

## 4. Discussion

Studies regarding the effectiveness of repeated massage on DOMS have been conducted after a single bout of stress and did not find significant differences from the control [8,13,14]. Because this is not the reality of sports in general, we investigated single and repeated effects of systematic massage adjusted for young swimmers’ training during one week on different outcomes to understand recovery in a practical scenario. 

We hypothesized that massage would improve swimmers’ perceptions shortly after intervention which could, in turn, improve their training abilities. In this sense, we found that the perception of the well-being of participants in both massage groups remained basically unchanged, whereas the participants in the CON reported a worsening of this perception at the end of the week. In fact, participants that received a massage had less chance of feeling tired from resistance training shortly after intervention at the end of the week. This may indicate a cumulative benefit from receiving massages throughout the week. However, we highlight that these effects were small and limited to self-reported perceptions.

Although massage has been shown to reduce acute pain [24], we did not find a significant difference between groups immediately after the intervention. In this study, the training load was monitored and increased but the swimmers were already used to the stimulus which did not result in the pain levels usually observed after experimental muscle-damaging exercise protocols and may explain the divergence from laboratory trials. However, we found evidence of the worsening perceptions of heaviness and pain at the main stages of the swim training for the massage groups. Although we did not investigate training quality, the fact that the swimmers were less tired after massage and felt pain during the sprints that did not remain after the swim training may suggest that the swimmers were able to train harder with no harm to recovery. A biopsy study conducted by Crane et al. shows that 10 min of massage reduced interleukin-6 and heat shock proteins and potentiated mitochondrial biogenesis signaling after 2.5 h [25], which may contribute to improving athletes’ abilities during swim training after the exertion from resistance training. 

This brings light to the debate regarding the management of training load in the sports scenario. For competitive swimmers, greater training loads were found to be associated with better performance [26], but the intensity during the week prior to injury was also significantly higher in young athletes [27]. Current knowledge on the training injury prevention paradox suggests that high loads are not the problem, but how they are prescribed. According to Gabbett, excessive and rapid increases in training loads are likely responsible for most non-contact, soft-tissue injuries [28]. The author also suggests that smarter training (greater amounts of short, high-intensity acceleration effort) may promote appropriate physical qualities to improve performance and protect athletes against injury [28]. Therefore, we suggest that, in addition to the adequate prescription of training loads, individual strategies should be implemented to aid systematic recovery throughout the season to improve training quality. 

Our second hypothesis, that the cumulative effects of massage between training sessions would result in positive repercussions for swimmers’ sprint and functional performance, was only partially confirmed, as we did not find significant statistical improvements on the tested variables after one week of superficial and deep massage compared to CON. On the other hand, although there is some hypothesis that too much recovery may hinder training adaptations [29], it was also not the case in our study because their performance remained unchanged. In this context, it is important to highlight that, given the training period included in the study in which swimmers were submitted to high training loads that promote structural and metabolic stress, they were not expected to perform their best times. In fact, if they were able to train harder without decreasing swim performance, massage can actually be included in the periodization as a recovery strategy to improve training abilities. Further assertions regarding the impact of recovery on performance can be made by investigating a longer period that includes the competitive phase. This investigation may be challenging due to training logistics and small sample sizes but reflects the actual scenario of athletic training and has not been investigated yet.

This study is not without limitations. We did not objectively assess swim quality after recovery or monitor an entire season, which leaves unanswered gaps. Thus, assumptions for other stages of training cannot be made. Therefore, investigating the behavior of recovery throughout the season is still relevant given the specificity of training principles. Findings of previous studies support the investigation of the application of massage for longer periods. Rapaport et al. [30] found that massage applied once a week for five weeks had the same effects as a single session, however, when applied twice a week for the same period, massage promoted changes in oxytocin, adrenocorticotropic hormone, and cortisol that lasted up to four days between sessions. Due to a small sample size, we did not perform a subgroup analysis for sex. According to Robelot et al., men and women show great differences in performance in speed tests [31], which was the criterion for performance in this study. Simulated sprints can also be considered a limitation once real competition may be more accurate to test athletic performance. Finally, the sample showed great competitive level diversity. Differences in physical and psychological conditions and the magnitude of improvement can influence the response to training and recovery; therefore, the results of this study cannot be extrapolated to older swimmers but bring light to the recovery characteristics of young athletes. 

## 5. Conclusions

Massage improved perceptions of well-being, heaviness, and pain over the training sessions but had no effects on sprint and functional performance outcomes after one week. The insertion of massage as a recovery strategy in the periodization did not reduce performance and, therefore, can be used for swimmers after resistance exercise. However, given its small effects, individual symptoms, and preference, training logistics, and athletic aims should be taken into account during decision-making.

## Figures and Tables

**Figure 1 ijerph-20-01677-f001:**
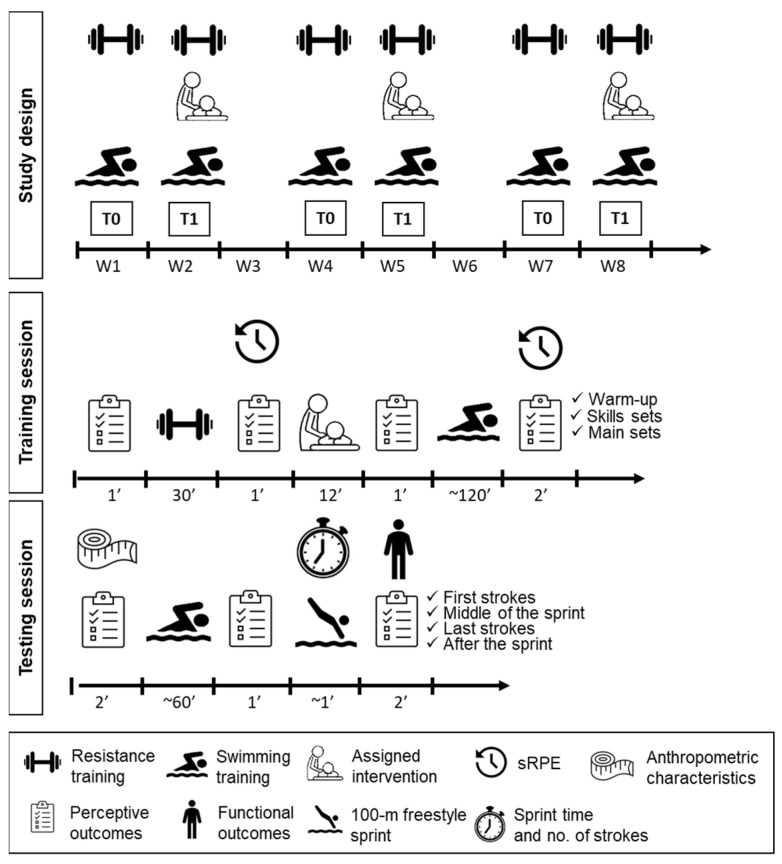
Study design and description of outcome assessments on training and testing sessions (T0 = training week; T1 = post-intervention week; W1–8 = weeks one to eight).

**Figure 2 ijerph-20-01677-f002:**
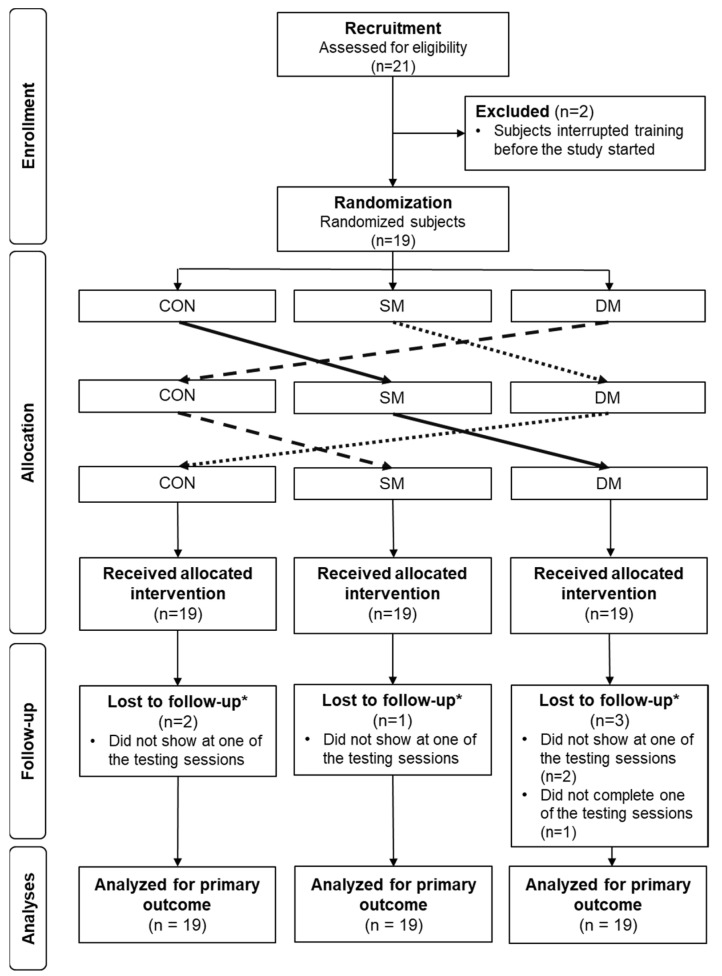
Flow diagram of the phases of the crossover randomized trial of three groups. (* Data from participants lost to follow-up were imputed by the last observation carried forward, CON = control, SM = superficial massage, DP = deep massage).

**Figure 3 ijerph-20-01677-f003:**
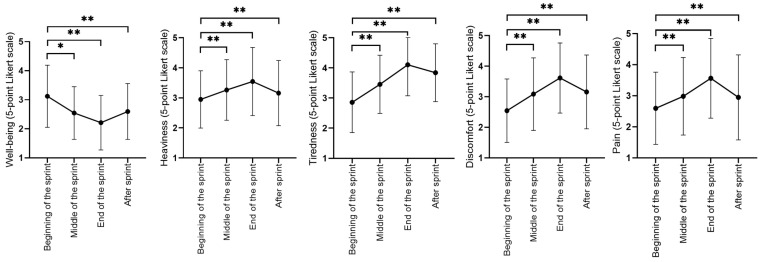
Behavior of perceptions during the 100-m freestyle sprint (* = *p* < 0.05, ** = *p* < 0.01).

**Figure 4 ijerph-20-01677-f004:**
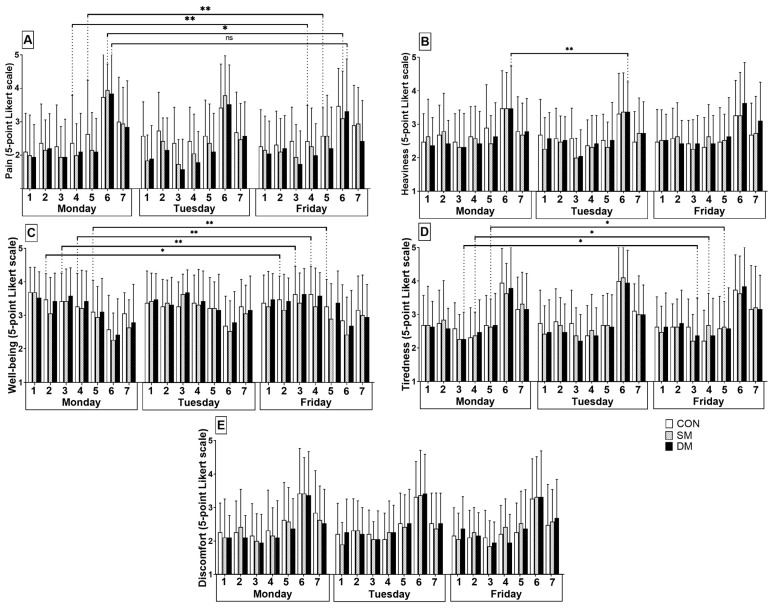
Behavior of perceptions of (**A**) pain, (**B**) heaviness, (**C**) well-being, (**D**) tiredness, and (**E**) discomfort during the training sessions of intervention days. (1 = before resistance training, 2 = after resistance training, 3 = after intervention, 4 = swim warm-up, 5 = after swim warm-up, 6 = main series, 7 = after swim training, * = *p* < 0.05, ** = *p* < 0.01, ns = non significant).

**Table 1 ijerph-20-01677-t001:** Anthropometric characteristics at baseline.

	Control (*n* = 17) Mean (SD)	Superficial Massage (*n* = 18) Mean (SD)	Deep Massage (*n* = 16) Mean (SD)
**Sex ^a^**			
Male	11 (64.7)	11 (61.1)	10 (62.5)
Female	6 (35.3)	7 (38.9)	6 (37.5)
**Age (years)**	13.71 (1.57)	13.61 (1.24)	14.06 (1.61)
**Body weight (kg)**	60.21 (13.07)	59.05 (11.32)	61.67 (12.57)
**Body height (cm)**	1.67 (0.09)	1.66 (0.09)	1.68 (0.10)
**BMI (kg/m^2^)**	21.43 (3.14)	21.23 (2.96)	21.74 (3.02)

^a^ Data presented as no. (%). Abbreviation: BMI = body mass index.

**Table 2 ijerph-20-01677-t002:** Generalized linear models for sprint and functional performance outcomes after one week of repeated massage (SM and DM) compared to control.

	Within Group Differences			Between Group Differences		
	CON Mean ± SD B (95% CI)	SM Mean ± SD B (95% CI)	DM Mean ± SD B (95% CI)	SM—CON B (95% CI)	DM—CON B (95% CI)	SM—DM B (95% CI)
*Squat jump* (cm)						
Baseline	26.80 ± 5.40	26.69 ± 5.55	27.78 ± 5.44	0.23 (−0.49, 0.96)	0.80 (−0.10, 1.71)	−0.57 (−1.42, 0.28)
At 1 wk	26.82 ± 5.40	27.40 ± 5.68	27.45 ± 5.36			
Baseline → 1 wk	0.02 (−1.00, 1.05)	0.70 (−0.33, 1.74)	−0.33 (−1.39, 0.72)			
*Bench press* (m/s)						
Baseline	1.32 ± 0.24	1.26 ± 0.25	1.27 ± 0.24	−0.04 (−0.11, 0.02)	−0.07 (−0.14, 0.00)	0.02 (−0.02, 0.08)
At 1 wk	1.33 ± 0.24	1.30 ± 0.25	1.23 ± 0.20			
Baseline → 1 wk	0.003 (−0.07, 0.08)	0.003 (−0.04, 0.11)	−0.03 (−0.11, 0.03)			
*Flexibility* (cm)						
Baseline	33.91 ± 8.04	34.85 ± 8.52	35.61 ± 8.20	0.87 (−0.79, 2.54)	1.03 (−0.74, 2.81)	−0.15 (−1.61, 1.30)
At 1 wk	33.73 ± 7.83	34.55 ± 8.27	34.11 ± 7.68			
Baseline → 1 wk	−0.18 (−2.19, 1.83)	−0.29 (−2.36, 1.76)	−1.49 (−3.57, 0.59)			
*Proprioception at 55°* (°)						
Baseline	9.36 ± 5.44	10.75 ± 6.32	9.75 ± 5.28	0.55 (−2.30, 3.42)	0.30 (−2.11, 2.73)	0.25 (−2.18, 2.68)
At 1 wk	11.08 ± 6.34	10.73 ± 6.32	11.28 ± 6.40			
Baseline → 1 wk	1.72 (−1.54, 4.98)	−0.02 (−3.37, 3.33)	1.53 (−1.78, 4.86)			
*Proprioception at 90°* (°)						
Baseline	6.41 ± 3.71	6.70 ± 3.98	5.76 ± 3.24	0.03 (−1.53, 1.59)	−0.80 (−2.59, 0.98)	0.84 (−0.97, 2.65)
At 1 wk	6.95 ± 3.71	6.71 ± 3.69	5.98 ± 3.08			
Baseline → 1 wk	0.53 (−1.66, 2.73)	0.005 (−2.21, 2.22)	0.22 (−1.71, 2.15)			
*Proprioception at 125°* (°)						
Baseline	9.01 ± 4.61	9.79 ± 5.13	9.46 ± 4.68	−0.20 (−2.37, 1.78)	0.10 (−1.93, 2.15)	−0.40 (−2.82, 2.02)
At 1 wk	9.96 ± 4.16	8.60 ± 3.69	9.70 ± 3.92			
Baseline → 1 wk	0.95 (−1.84, 3.75)	−1.18 (−3.97, 1.60)	0.24 (−2.60, 3.08)			
*Total time* (s)						
Baseline	68.91 ± 6.14	68.82 ± 6.27	69.87 ± 6.04	−0.01 (−1.40, 1.37)	0.23 (−1.24, 1.71)	−0.25 (−1.95, 1.45)
At 1 wk	68.88 ± 5.23	68.94 ± 5.38	68.40 ± 5.04			
Baseline → 1 wk	−0.03 (−1.99, 1.92)	−0.12 (−1.82, 2.08)	−1.46 (−3.44, 0.51)			
*FINA points*						
Baseline	872.46 ± 151.24	876.26 ± 156.30	857.62 ± 144.20	−0.16 (−42.71, 43.03)	1.17 (−51.39, 53.73)	−1.01 (−44.22, 42.19)
At 1 wk	872.79 ± 125.30	869.33 ± 128.42	890.28 ± 124.00			
Baseline → 1 wk	0.33 (−60.21, 60.87)	−6.92 (−67.59, 53.74)	32.66 (−27.50, 92.83)			
*Swimming velocity*						
Baseline	1.35 ± 0.12	1.36 ± 0.12	1.35 ± 0.12	−0.03 (−0.10, 0.03)	−0.02 (−0.09, 0.04)	−0.01 (−0.06, 0.05)
At 1 wk	1.42 ± 0.16	1.35 ± 0.16	1.37 ± 0.16			
Baseline → 1 wk	0.06 (−0.01, 0.14)	−0.01 (−0.09, 0.06)	0.01 (−0.06, 0.09)			
*Stroke frequency*						
Baseline	1.40 ± 0.12	1.40 ± 0.12	1.39 ± 0.12	−0.01 (−0.06, 0.05)	−0.03 (−0.10, 0.04)	0.02 (−0.04, 0.09)
At 1 wk	1.46 ± 0.16	1.46 ± 0.16	1.41 ± 0.12			
Baseline → 1 wk	0.06 (−0.02, 0.14)	0.06 (−0.02, 0.14)	0.02 (−0.06, 0.10)			
*Distance per stroke*						
Baseline	0.97 ± 0.12	0.98 ± 0.12	0.98 ± 0.12	−0.01 (−0.04, 0.02)	0.00 (−0.02, 0.03)	−0.01 (−0.04, 0.02)
At 1 wk	0.97 ± 0.12	0.93 ± 0.12	0.97 ± 0.12			
Baseline → 1 wk	0.00 (−0.04, 0.04)	−0.04 (−0.08, 0.00)	−0.00 (−0.05, 0.03)			
*Stroke index*						
Baseline	1.32 ± 0.28	1.34 ± 0.29	1.34 ± 0.28	−0.05 (−0.15, 0.04)	−0.02 (−0.11, 0.06)	−0.03 (−0.12, 0.06)
At 1 wk	1.40 ± 0.28	1.27 ± 0.29	1.33 ± 0.28			
Baseline → 1 wk	0.07 (−0.04, 0.19)	−0.07 (−0.18, 0.03)	−0.01 (−0.11, 0.11)			

Abbreviations: CON = control; DM = deep massage; SM = superficial massage; wk = week, SD = standard deviation, CI = confidence interval.

## Data Availability

The data that support the findings of this study are openly available in Mendeley Data at http://doi.org/10.17632/d88r6svvds.3, reference number [32].

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
