# Peer review of "Repeated Massage Improves Swimmers’ Perceptions during Training Sessions but Not Sprint and Functional Performance: A Randomized Controlled Trial"

_ijerph, 2023, doi:10.3390/ijerph20031677_

Round 1

Reviewer 1 Report

the article presented is relevant to understand the effect of massage on the recovery of swimmers.
Swimmers exert a great deal of effort during long hours of training and recovery is essential for their performance in competition.
The protocol designed is based on existing evidence and as indicated in the limitations will have to be performed on a larger sample.

Author Response

The article presented is relevant to understand the effect of massage on the recovery of swimmers. Swimmers exert a great deal of effort during long hours of training and recovery is essential for their performance in competition. The protocol designed is based on existing evidence and as indicated in the limitations will have to be performed on a larger sample.

Response: We appreciate the time to read and consider our study. We agree that to extrapolate these results to competitive athletes a larger study would be of great importance due to possible differences that could result from differences in sex, age and even in competitive level. We would also like to highlight that because of the crossover design our study still has a larger sample size than many study with athletes and therefore can bring important initial evidence that contributes to the research in the sports field.

Reviewer 2 Report

Dear Authors,

It would be appropriate to add all measured parameters to the table.The authors mainly refer to changes in the perception of well-being, heaviness, tiredness and pain, while these parameters are not presented in the table.  

Author Response

Dear Authors,

It would be appropriate to add all measured parameters to the table. The authors mainly refer to changes in the perception of well-being, heaviness, tiredness and pain, while these parameters are not presented in the table.

Response: We appreciate the reviewer’s suggestion to improve our results section. We made the option not to report the perceptions in table 2 because the amount of information compared for the functional and performance outcomes is different than for the perceptive outcomes. Although all outcomes were assessed before and after the intervention week, the perceptive outcomes we are interested in are related to the sprint (Beginning of the sprint, Middle of the sprint, End of the sprint, After the sprint) and therefore have more comparations than the other outcomes that we only compare before and after the intervention week. Adding the perceptive outcomes to this table, in addition to making it longer than it already is (which was highlighted by the third reviewer) could confound the reader.

The changes in perceptions that we refer in the results are from the training sessions, which have also many time-points that are also different from the others (before resistance training; after resistance training; after intervention; swim warm-up; after swim warm-up; main series; and after swim training). In this sense, because there are too many comparations and the differences are punctual we believe that another table would not be intuitive to read. To overcome this issue, we wrote a different text that we believe would be clearer, and therefore we suggest that the reviewer consider this new approach.

Otherwise, we also propose two new figures to facilitate reading. Figure 3 shows time effects and changes in perceptions during the sprint, while figure 4 shows description of the perceptions during the training sessions throughout the study weeks and the time*group effects that are previously described in the text.

Reviewer 3 Report

See attached

Author Response

General

The current manuscript looks to compare repeated massage on several measurements during a swimmers training. This study was interesting because of it’s uses two different types of massage, as well as a control, over a period of a training cycle. Previous studies have primarily involved single bouts of massages and not as many variables. The study found that swimmers’ perceptions of well-being, heaviness, and pain over training sessions was improved during massage versus the control. While the manuscript is well written I do feel some revisions need to be made before publication.

COMMENTS

General: There are several areas in the manuscript that were difficult to read because of the lack of using an oxford comma. While the use of an oxford comma is not required, I believe it would improve clarity of sentences.

Response: We would like to thank the reviewer’s suggestion. Indeed the use of the oxford comma can improve the understanding of some sentences of our word, mainly in the methods section. In this new version of the manuscript we adopted the used the oxford comma.

Page 1, Line 1-3: I struggled with understanding the study based on the title. The use of ‘over’ just makes it read very confusing. Maybe a switch to ‘during’ would improve clarity.

Response: The reviewer can find these changes in the new version of the manuscript

Abstract

Page 1, Line 10-12: The first sentence of the abstract should be split into two sentences. To confusing to read as written.

Page 1, Line 16: Should read ‘swim training session and…’

Page 1, Line 18: I believe functional should be function, or should be “functional something”. You also use the term functional throughout the manuscript in the same way.

Page 1, Line 23-24: You use the abbreviations SM, DM, and CON without explaining them first. Abbreviations should only be used after spelling the words out, even in the abstract.

Response: We would like to thank the reviewer for the suggestions in the abstract. The reviewer can find these changes in the new version of the manuscript. We would also like to clarify that throughout the text we use the term ‘functional outcomes’ but we indeed missed this word in the abstract, and therefore this was adjusted.

Introduction

General

Page 1, Line 32: Should be ‘coaches’, not ‘coach’

Page 1, Line 33: Should be ‘follow’, not ‘followed’

Response: The reviewer can find these changes in the new version of the manuscript

Page 2, Lines 47-49: The sentence beginning ‘Swimmers have reported…’. You need to reference the studies were swimmers reported one session of massage is not enough.

Response: We would like to thank the reviewer for spotting this inconsistency, we added the reference to this new version of the manuscript.

Methods

Page 4, Line 112: In your intervention session, you need to describe what the control group did during the time that the other two groups were getting a massage.

Response: The sentence describing the control was in another paragraph separate from the massage interventions, but this is indeed better altogether. In this version the description of the control is on the same paragraph.

Results

Results are very well written, and the tables do a good job of showing all the data. Tables 2 was long and covered 3 pages. Probably not your fault but work with the publisher to get a table that fits on one page.

Response: We made few editions on table 2 and it fits one page.

Discussion

Your discussion was thorough and covered all the relevant comparisons and potential reasons for results.